# Sex-disaggregated data along the gendered health pathways: A review and analysis of global data on hypertension, diabetes, HIV, and AIDS

**Alessandro Feraldi**[1], **Virginia Zarulli**[2], **Kent Buse**[3,4], **Sarah Hawkes**[3,4], **Angela Y. Chang**[5,6]*

**1** Department of Methods and Model for Economics, Territory, and Finance, Sapienza University of Rome, Rome, Italy, **2** Department of Statistical Sciences, University of Padova, Padova, Italy, **3** Global 50/50, Cambridge, United Kingdom, **4** Department of Global Population Health, Monash University Malaysia, Kuala Lumpur, Malaysia, **5** Danish Centre for Health Economics, University of Southern Denmark, Odense, Denmark, **6** Danish Institute for Advanced Study, University of Southern Denmark, Odense, Denmark

* achang@health.sdu.dk

## Abstract

### Background

Health data disaggregated by sex is vital for identifying the distribution of illness, and assessing risk exposures, service access, and utilization. Disaggregating data along a health pathway, i.e., the measurable continuum from risk factor exposure to final health outcome (death), and including disease prevalence and a three-step care cascade (diagnosis, treatment, and control), has the potential to provide a holistic and systematic source of information on sex- and gender-based health inequities and identify opportunities for more tailored interventions to reduce those inequities.

### Methods and findings

We collected sex- and age-disaggregated data along the health pathway. We searched for papers using global datasets on the sex-disaggregated care cascade for eight major conditions and identified cascade data for only three conditions: hypertension, diabetes, and HIV and AIDS. For each condition, we collected risk factor prevalence, disease prevalence, cascade progression, and death rates. We assessed the sex difference for all steps along the pathway and interpreted inequities through a lens of gender analysis. Sex-disaggregated data on risk factors, disease prevalence, and mortality were found for all three conditions across 204 countries. Sex-disaggregated care cascades for hypertension, diabetes, and HIV and AIDS were found only for 200, 39, and 76 countries, respectively. Significant sex differences were found in each step along the pathways. In many countries, males exhibited higher disease prevalence and death rates than females, while in some countries, they also reported lower rates

**Data availability statement:** Data on prevalence and death rates are available at the Global Burden of Disease Collaborative Network. https://www.healthdata.org/research-analysis/gbd. Data on risk factors are available are available upon request from Institute for Health Metrics and Evaluation, https://www.healthdata.org/. Data on hypertension cascade of care are available at the NCD-RisC database, available at https://www.ncdrisc.org/data-downloads-hypertension.html. Data on diabetes cascade of care are available at the World Health Organization's STEPWise Approach to NCD Risk Factor Surveillance (STEPS) survey, available at https://www.who.int/teams/noncommunicable-diseases/surveillance/systems-tools/steps/data-collection-tools. Data on HIV cascade are available at the Joint United Nations Programme on HIV and AIDS (UNAIDS), https://aidsinfo.unaids.org/.

**Funding:** This work was supported by the Bill & Melinda Gates Foundation (grant INV-030827) to Global Health 50/50. The funders had no role in study design, data collection and analysis, decision to publish, or preparation of the manuscript.

**Competing interests:** I have read the journal's policy and the authors of this manuscript have the following competing interest: KB is on the board of and chairs the policy committee of the World Obesity Federation which receives industry funding. KB and SH are co-CEOs of Global Health 50/50 - - an NGO which receives grants from Gates Foundation to work on issues relating to gender and global health. SH is co-Chair of the Lancet Commission on Gender and Global Health which received funds from the Wellcome Trust, Ford Foundation and GH5050 for its work. SH is a Board Member of the Elsevier Inclusion and Diversity Board.

**Abbreviation:** GBD, Global Burden of Disease; IHME, Institute for Health Metrics and Evaluation; NCD-RisC, NCD Risk Factor Collaboration; BMI, Body Mass Index; WHO, World Health Organization

of healthcare seeking, diagnosis, and treatment adherence. Smoking prevalence was higher among males in most countries, whereas prevalence of obesity and unsafe sex were higher in females in most countries.

## Conclusions

Findings support the increasing need to develop strategies that encourage greater male participation in preventive and healthcare service and underscore the importance of sex-disaggregated data in understanding health inequities and guiding gender-responsive interventions at different points along the pathway. Despite limitations in data availability and completeness, this study elucidates the need for more comprehensive and harmonized datasets for these and other conditions to monitor sex differences and implement sex-/gender-responsive interventions along the health pathway.

## Author summary

### Why was this study done?

- Despite strong evidence that sex and gender influence health outcomes, many health policies lack sex-specific approaches.

- Prior research on health trends presents results separated by sex, limiting discussions on sex differences and geographic coverage.

### What did the researchers do and find?

- We reviewed global datasets and published papers on sex-disaggregated care cascades for eight major health conditions and analyzed sex differences along the health pathway.

- Findings revealed that males face a double burden of higher prevalence of risk factors and diseases while also experiencing lower access to diagnosis and treatment.

- The study shows the lack of sex-disaggregated care cascade data compared to risk exposure and health outcomes, creating gaps in understanding health inequities.

### What do these findings mean?

- The results highlight the need for strategies that encourage greater male participation in preventive and healthcare services.

- Findings underscore the importance of comprehensive sex-disaggregated data to better understand health inequities and inform gender-responsive interventions.

- More standardized and harmonized global datasets are needed to monitor sex differences and guide equitable healthcare policies.

PLOS Medicine

## Introduction

Sex and gender significantly shape health outcomes including through their individual or combined influence on a range of factors, such as rates of exposure to risk factors and environments, individual bodily responses to risk exposure, and patterns of health service use [1]. For example, gender, as a social construction, influences how men and women have different exposure to health-harming products such as tobacco, alcohol, and poor nutrition. Sex plays an important role in health outcomes, including by determining physiological responses to gendered risk exposures (e.g., male and female bodies metabolize alcohol differently), while gender influences healthcare access and the quality of care received from health services [2,3].

Understanding differences in risk exposure, health service use, and health outcomes by sex and gender identity (i.e., whether someone identifies as a woman, man or another gender identity) is vital to the identification of interventions which can effectively reduce health inequities, e.g., ensuring that interventions are tailored to the specific needs of women, men, and gender-diverse individuals. However, sex and gender identity are often confused and conflated in health surveys [1]. As a result, analysis of survey data is challenging—were researchers reporting on sex or gender identity? Does the original language of the survey have different terms to distinguish between sex and gender identity? Additionally, very few surveys outside of a few specific country contexts report on gender identity beyond a simple binary (woman/man) [4]. In this study, we report on data from datasets and surveys as being sex-disaggregated, but interpret any observed differences through the lens of gender analysis—which enables identification of modifiable factors for sex- and gender-responsive interventions.

We analyzed global datasets to construct what we call a health pathway—from risk exposure to mortality (illustrated in Fig 1). The health pathway approach can be applied to improve our understanding of where differences (if any) are seen between males/females, and identify opportunities for tailored interventions to reduce inequities.

To illustrate the health pathway concept, we present sex-disaggregated data along each step for three conditions with available global data. Disease modeling groups, such as the Institute for Health Metrics and Evaluation (IHME) and NCD Risk Factor Collaboration (NCD-RisC) [5], provide sex- and age-specific estimates for some steps of the health pathway—particularly, risk factor prevalence, disease prevalence, and mortality. The "care cascade" is a framework commonly

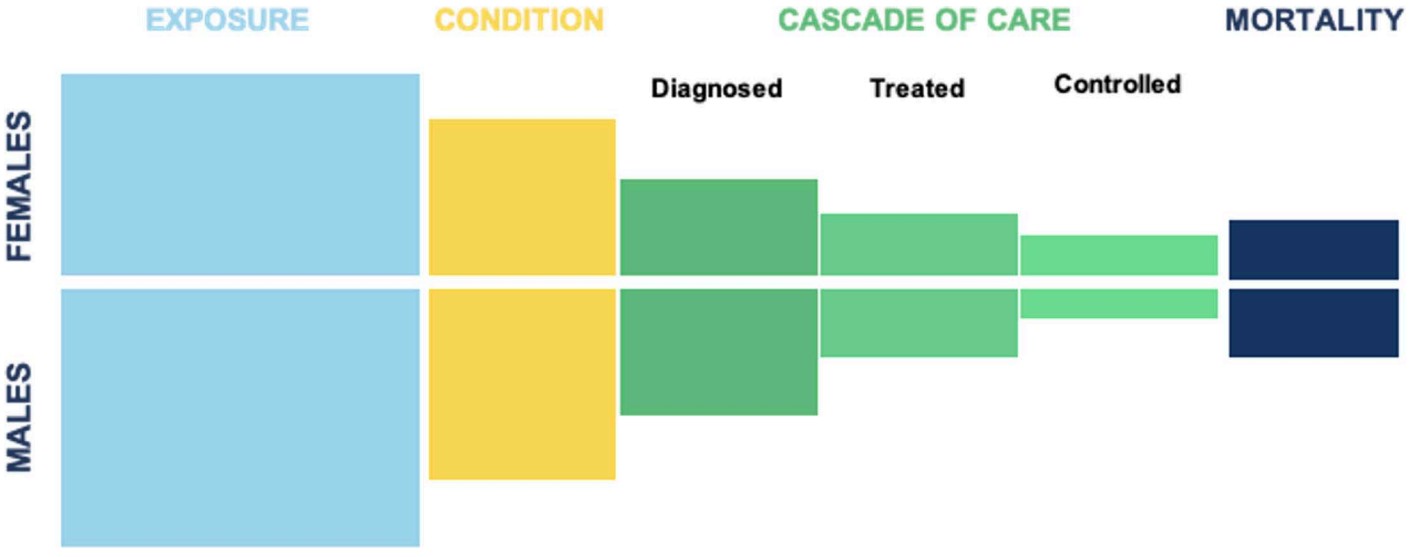

**Fig 1. Illustration of the health pathway.**

used to assess how people with a disease/condition are retained through the sequential stages of care (diagnosis, treatment, disease control), emphasizing its role in tracking patient progress and enhancing access to and quality of care [6–8]. While some regional and national studies offer sex-disaggregated data on the care cascade for specific conditions, globally comparable data remains scarce [9–12]. We conducted a scoping review to identify global sex-disaggregated cascade datasets for eight common conditions, and found data on only three: hypertension, diabetes, and HIV and AIDS. Among the global studies that have analyzed these datasets, none have specifically examined sex differences in the care cascade [6,13–17]. Our study addresses this data gap by constructing a pathway from exposure to outcome, including the care cascade, examining sex differences at each stage of the pathway, for three conditions.

## Methods

### Scoping review for the care cascade data and selection of diseases

Recognizing the scarcity of data on the care cascade compared to risk exposure and health outcomes, we searched for published papers using global datasets on the sex-disaggregated care cascade for eight major conditions and causes: hypertension, diabetes, HIV and AIDS, tuberculosis, dementia, depression, chronic obstructive pulmonary disease, and lung cancer. We looked for global-level datasets that report the cascade steps by sex for more than five countries. More specifically, we searched on MEDLINE (via PubMed) for articles published sex- or gender-disaggregated data on each step of the health pathway until March 26th, 2024. Our search criteria included specific terms: ("Care cascade" OR "cascade of care" OR ((diagnosis OR detection) AND (medication OR treatment) AND (control OR aware*))) AND ((sex-specific OR sex-disaggregated OR "by sex") OR (gender-specific OR gender-disaggregated OR "by gender")) AND (trend* OR global OR worldwide). We made no language restrictions. Through the scoping review we identified three key issues related to data gaps and availability: (1) a general lack of globally comparable cascade data, (2) among the available cascade data, a frequent absence of sex and/or age disaggregation; (3) even when cascade data are disaggregated by sex, studies often fail to address or discuss sex-related differences. The review identified global datasets meeting these criteria only for hypertension, diabetes, and HIV and AIDS. These three conditions were selected for further analysis, shaping the subsequent research methods.

### Data sources

**Risk factors, disease prevalence, and death rates.** Data on risk factors, disease prevalence, and death rates were obtained from the Global Burden of Disease (GBD) dataset [5]. Although the GBD database presents some limitations, it remains the most reliable resource for comprehensive global data on risk factors and disease prevalence. While in-country data are valuable, the GBD data are standardized—meaning it is possible to undertake comparisons across countries [18–20]. To align with the care cascade data and ensure consistency in analyses, the study relied on 2019 data for risk factors, prevalence, and death rates from the GBD 2021 release.

For diabetes and HIV and AIDS, we identified the risk factors that globally had the highest risk factor attributions to disease-specific deaths in the 2019 GBD data [5], and for which sex- and age-disaggregated prevalence data were available. For diabetes, these were: fasting plasma glucose (estimated as a continuous risk factor reflecting mean fasting plasma glucose) of at least 7 mmol/L (126 mg/dL) or currently treated with drugs or insulin (measured as proportion of population (%)); overweight (% of people with BMI above 25 kg/m$^2$); obesity (% of people with BMI above 30 kg/m$^2$); smoking of any tobacco product in the last 30 days (%); and low physical activity (average weekly physical activity [at work, home, transport related, and recreational] measured in Metabolic Equivalent of Task minutes per week). For HIV and AIDS, we included drug use (estimates of prevalence of drug use disorders, %), unsafe sex (estimated as the proportion of HIV and AIDS that is due to unsafe sex, %), and intimate partner violence (defined as lifetime prevalence of sexual and/or physical violence perpetrated by an intimate partner, %, reported only for women). Since hypertension (high systolic blood pressure) is categorized as a risk factor for cardiovascular diseases rather than a cause in the GBD, we used risk

factors with the highest attributions to cardiovascular diseases instead. These were high sodium intake (estimated as a continuous risk factor reflecting mean grams of sodium intake, grams per day), high fasting plasma glucose (%), over-weight (%), obesity (%), and smoking (%). Data on risk factors, disease prevalence, and death rates were available for 204 countries (see **Table A** in S1 Text).

**Cascade of care.** The care cascade framework includes the following three steps: (1) Diagnosed (step 1): the proportion of individuals diagnosed with the condition, among the total population estimated to be living with the condition; (2) Treated (step 2): the proportion of individuals with the condition receiving treatment; and (3) Controlled (step 3): proportion of individuals with the condition who are effectively controlled through treatment. We also calculated the conditional steps: proportion of people in step 1 who reached step 2 and proportion of people in step 2 who reached step 3.

Data on the hypertension care cascade came from NCD-RisC, which included sex-disaggregated cascade data across 200 countries and territories (see **Table A** in S1 Text), from 1990 to 2019, in 5-year age groups, from age 30–34 to 75–79. Hypertension was defined as having systolic blood pressure 140 mm Hg or greater, diastolic blood pressure 90 mm Hg or greater, or taking medication for hypertension [6,21]. For a detailed description of the dataset, its sources and inclusion criteria see [6] and [22]. For diabetes, data came from World Health Organization's STEPWise Approach to NCD Risk Factor Surveillance (STEPS) survey, which focuses on low- and middle-income countries [23]. We restricted our analysis to surveys between 2013 and 2020 and selected the latest year available for countries, which included only 39 countries, by age groups 30–44, 45–59, and 60–79. For HIV and AIDS, we used data from the Joint United Nations Programme on HIV and AIDS (UNAIDS). Data were available over the period 2015–2021, for 174 countries (with availability varying by country), and for ages 15+ as one age group. To ensure consistency and avoid potential biases from COVID-19-related disruptions, the study focused on data from 2019 for hypertension and HIV and AIDS care cascades. Hypertension cascade data for 2019 were available for 200 countries, while HIV and AIDS care cascade data were complete for 76 countries. For diabetes, given the limited availability of care cascade data across years and countries, the study pooled data from 2013 to 2020, yielding information for 39 countries.

## Statistical analysis

We examined the sex differences along the health pathway, including risk factor exposure, disease prevalence, stages of the care cascade, and death rates for each condition. We identified significant sex differences in health pathway steps by comparing the 95% confidence intervals between females and males, considering steps with non-overlapping intervals as statistically different. By treating the male and female estimates as independent, this method provides strong statistical power by adopting a stringent significance threshold (i.e., $0.05 \times 0.05 = 0.0025$, resulting in a significance level of 0.25%) [24]. As confidence intervals for diabetes care cascade were unavailable, we calculated exact 95% binomial confidence intervals.

For regional comparisons, we used country classifications based either on 2023 World Bank income groups or on the World Bank's regional classification, except for Tokelau, Niue, and Cook Islands, for which no classification was available. Sixty-seven countries were classified as high-income, 54 as upper-middle-income, 54 as lower-middle-income, and 26 as low-income (total 201). In terms of regions, 52 countries were in Europe and Central Asia, 8 in South Asia, 44 in sub-Saharan Africa, 22 in the Middle East and North Africa, 38 in Latin America and the Caribbean, 34 in East Asia and the Pacific, and 3 in North America.

## Results

Fig 2A–2C illustrates the distribution of countries with observed sex differences in various stages of the health pathway for the three conditions. We present each pathway step on the x-axis and age groups on the y-axis. For each pathway step-age combination, the figures show whether more countries report males had significantly higher proportions than females or vice versa. Notice that for hypertension, risk factors, prevalence, cascade, and death rates refer to 204 countries and

**Panel A**

**Hypertension Health Pathways \***

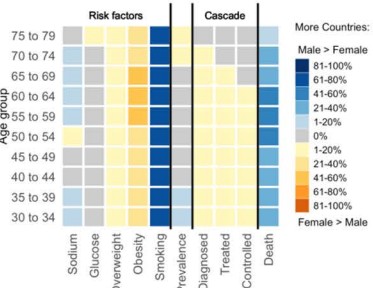

\* Risk factors, disease prevalence, and death rates are available for 204 countries and 5-years age groups. Hypertension care cascade data are available for 200 countries and 5-years age groups.

**Panel B**

**Diabetes Health Pathways \***

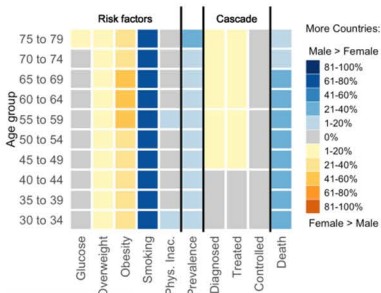

\* Risk factors, disease prevalence, and death rates are available for 204 countries and 5-years age groups. Diabetes care cascade data are available only for 39 countries and age groups 30-44, 45-59, and 60-79. Phys. Inac.: Physical inactivity.

**Panel C**

**HIV and AIDS Health Pathways \***

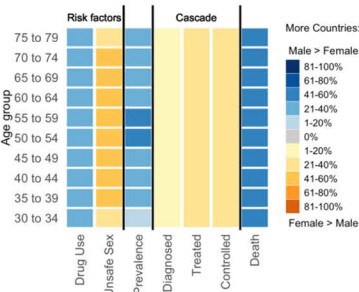

\* Risk factors, disease prevalence, and death rates are available for 204 countries and 5-years age groups. HIV and AIDS care cascade data are available only for 76 countries and one age group (15+).

**Fig 2. Sex differences (significant when non-overlapping confidence intervals of estimates between females and males) in global health pathways of hypertension, diabetes, and HIV and AIDS.**

5-years age groups, but care cascade data refer to 200 countries and similar age groups. For diabetes and HIV and AIDS, risk factors, prevalence, and death rates refer to 204 countries and 5-years age, whereas care cascade data were available for fewer countries and more aggregated age groups. Differences in the number of countries included in each step are specified in the footnotes, and differences in age groups are reflected by aggregated cells across age groups (Panels B and C). The same figures stratified by income groups and regions are reported in the **Figs A–F** in S1 Text, and percentage of countries with significant sex differences by income groups and regions are in **Fig G–M** in S1 Text. Significant sex differences in hypertension, diabetes, and HIV and AIDS health pathways across countries, regions, and income groups are reported in Tables 1–3, respectively.

**Table 1. Significant sex difference in hypertension health pathway.**

| | | Males > Females | Females > Males |
|---|---|---|---|
| **Risk factors** | **High Sodium Intake** | In 10 countries in some age groups (mostly in Europe and Central Asia, and high-income countries). Significant in all age groups in India. | In 7 countries at ages 50–54 (mostly in sub-Saharan Africa, and low- and lower-middle-income countries). Significant in all age groups in Kenya. |
| | **High Fasting Plasma Glucose** | None. | Only in Egypt at ages 75–79. |
| | **Overweight** | In 18 countries in some age groups (mostly in Europe and Central Asia, and high-income countries and at ages 30–39). Significant in more than half of the age groups in the UK and New Zealand. | In 58 countries in some age groups (mostly in sub-Saharan Africa, and middle-income countries). Significant in more than half of the age groups in 18 countries. |
| | **Obesity** | Only in China at ages 30–34 and in Malawi at ages 70–79. | In 130 countries in some age groups (mostly in sub-Saharan Africa, and low- and middle-income countries). Significant in more than half of the age groups in 63 countries. |
| | **Smoking** | In 176 countries in some age groups (mostly in sub-Saharan Africa and Europe and Central Asia, and high- and middle-income countries). Significant in more than half of the age groups in 138 countries. | Only in Bhutan at ages 65–69. |
| **Prevalence** | | In 8 countries in some age groups between 30 and 54 (mostly in Europe and Central Asia, and upper-middle-income countries). Highlighted*: UK. | Only in India at ages 70–79. |
| **Cascade Of Care** | **Diagnosis** | None. | In 8 countries in some age groups (mostly in sub-Saharan Africa, all middle-income countries). Highlighted*: Iran, Peru. |
| | **Treatment** | None. | In 4 countries in some age groups (mostly in Latin America and the Caribbean, all middle-income countries). Highlighted*: Iran, Peru. |
| | **Control** | None. | Only in Iran, Uzbekistan, and Peru*. |
| **Death** | | In 107 countries in some age groups (mostly in Europe and Central Asia, and high- and upper-middle-income countries). Significant in more than half of the age groups in 67 countries. | Only in the United Arab Emirates at ages 70–79. |

* Highlighted countries had significant differences in more than half of the age groups included in the analysis.

Table 2. Significant sex difference in diabetes health pathway.

| | | Males > Females | Females > Males |
|---|---|---|---|
| **Risk Factors** | **High Fasting Plasma Glucose** | None. | Only in Egypt at ages 75–79. |
| | **Overweight** | In 18 countries in some age groups (mostly in Europe and Central Asia, and high-income countries and at ages 30–39). Significant in more than half of the age groups in the UK and New Zealand. | In 58 countries in some age groups (mostly in sub-Saharan Africa, and middle-income countries). Significant in more than half of the age groups in 18 countries. |
| | **Obesity** | Only in China at ages 30–34 and in Malawi at ages 70–79. | In 130 countries in some age groups (mostly in sub-Saharan Africa, and low- and middle-income countries). Significant in more than half of the age groups in 63 countries. |
| | **Smoking** | In 176 countries in some age groups (mostly in sub-Saharan Africa and Europe and Central Asia, and high- and middle-income countries). Significant in more than half of the age groups in 138 countries. | Only in Bhutan at ages 65–69. |
| | **Low Physical Activity** | Only in Jamaica, Samoa, and Trinidad and Tobago in some age groups. | None. |
| **Prevalence** | | In 61 countries in some age groups (mostly in Europe and Central Asia, and sub-Saharan Africa, and high-income countries). Significant in more than half of the age groups in 23 countries. | In 10 countries in some age groups (mostly in Latin America and the Caribbean, and middle-income countries). Highlighted*: Azerbaijan, Haiti, Lao PDR, and Mauritania. |
| **Cascade Of Care** | **Diagnosis** | None. | Only in Cape Verde at ages 45–79. |
| | **Treatment** | None. | Only in Cape Verde at ages 45–79. |
| | **Control** | None. | None. |
| **Death** | | In 100 countries in some age groups (mostly in Europe and Central Asia, and high- and upper-middle-income countries). Significant in more than half of the age groups in 53 countries. | In 9 countries in less than half of the age groups (mostly in Latin America and the Caribbean, and upper-middle-income countries). Highlighted*: Afghanistan and Guatemala. |

* Highlighted countries had significant differences in more than half of the age groups in the analysis.

## Hypertension

**Risk factor exposure.** Male smoking rates significantly surpass those of females in 176 (86%) countries, with exceptions noted in Bhutan (ages 65–69) where the trend reverses. Obesity rates are higher in females across 130 countries (64%), except in China (ages 30–39) and Malawi (ages 70–79) where male obesity prevails. Overweight prevalence appears similar between the sexes in most countries, though in 18 countries (9%) males exhibit significantly higher rates and in 58 countries (29%) females do. High fasting plasma glucose levels are similar between sexes, except for Egypt where higher rates in females are observed. High sodium intake typically does not differ between sexes, but significant differences are found in 10 countries (5%) with higher rates in males, and in 7 countries (4%) higher in females.

**Disease prevalence.** Globally males and females had similar prevalence of hypertension, except for eight countries (4%) where male prevalence was significantly higher than female prevalence in some age groups. India is the only country with higher female prevalence at ages 70–79.

**Care cascade.** Data was available only 200 countries (63 high-income, 54 upper-middle-income, 54 lower-middle-income, and 26 low-income countries; 50 countries in Europe and Central Asia, 8 in South Asia, 44 in sub-Saharan Africa, 22 in Middle East and North Africa, 37 in Latin America and the Caribbean, 33 in East Asia and the Pacific, and

**Table 3. Significant sex difference in HIV and AIDS health pathway.**

| | | Males > Females | Females > Males |
|---|---|---|---|
| **Risk Factors** | **Drug Use** | In 139 countries in some age groups (mostly in Europe and Central Asia, and in sub-Saharan Africa, and high-income countries). In more than half of the age groups in 41 countries. | Only in Afghanistan, Belarus, China, Iceland, and Syria, in some age groups. |
| | **Unsafe Sex** | Only in Nepal at ages 30–34. | In 113 countries in some age groups (mostly in Europe and Central Asia and East Asia and the Pacific, and high- and upper-middle-income countries). Significant in more than half of the age groups in 82 countries. |
| **Prevalence** | | In 114 countries in some age groups (mostly in Europe and Central Asia, and Latin America and the Caribbean, and upper-middle- and high-income countries). Significant in more than half of the age groups in 68 countries. | In 28 countries in some age groups (mostly in sub-Saharan Africa, and lower-middle-income countries). Significant in more than half of the age groups in Tajikistan and South Africa. |
| **Cascade Of Care** | **Diagnosis** | None. | In 9 countries at all ages (mostly in Europe and Central Asia and sub-Saharan Africa, and lower-middle-income countries). |
| | **Treatment** | Only in Lebanon at all ages. | In 20 countries at all ages (mostly in Europe and Central Asia and sub-Saharan Africa, and lower-middle-income countries). |
| | **Control** | Only in Lebanon at all ages. | In 21 countries at all ages (mostly in Europe and Central Asia and sub-Saharan Africa, and lower-middle-income countries). |
| **Death** | | In 131 countries in some age groups (mostly in Europe and Central Asia, and Latin America and the Caribbean, and upper-middle- and high-income countries). Significant in more than half of the age groups in 113 countries. | In 25 countries in some age groups (mostly in the Middle East and North Africa, and high-income countries). Significant in more than half of the age groups in 6 countries (Greenland, Jordan, Kiribati, Lithuania, Qatar, and Tajikistan). |

* Highlighted countries had significant differences in more than half of the age groups in the analysis.

3 countries in North America). No significant sex differences were found in the progression along the cascade, although with some exceptions. Female diagnosis rates were significantly higher at ages 35–44 in seven countries (4%). Fewer countries showed significant differences at ages 45–74, and none at 75+. Female treatment rates were higher at 35–39 in four countries (2%). Fewer differences at 40–69, and none at 70+. Iran, Peru, and Uzbekistan females had higher rates of having their hypertension controlled at ages 30–39, with Peru showing significant sex differences until age 64. Diagnosis and treatment rates were higher among females in eight countries (4%) and four countries (2%), respectively.

**Death rate.** In 107 countries (53%), hypertension death rates were higher among males, whereas death rates were higher among females in the United Arab Emirates at ages 70–79.

### Diabetes

**Risk factor exposure.** The diabetes health pathway includes most of the same risk factors as hypertension, but with low physical activity instead of high sodium intake. Prevalence of low physical activity was similar among females and males, except in Jamaica, Samoa, and Trinidad and Tobago, where low physical activity was significantly higher among males in some age groups between 30 and 64.

**Disease prevalence.** Overall, in most countries there were no significant sex differences. Prevalence was significantly higher among males in 61 countries (30%) and among females in 10 countries (5%).

**Care cascade.** Data for diabetes was available only for 39 countries (4 high-income, 10 upper-middle-income, 20 lower-middle-income, and 5 low-income countries; 8 countries in Europe and Central Asia, 4 in South Asia, 11 in sub-Saharan Africa, 6 in Middle East and North Africa, 2 in Latin America and the Caribbean, 8 in East Asia and the Pacific, and no countries in North America) and only for ages 30–44, 45–59, and 60–79. Additionally, no confidence intervals were provided, therefore sex comparisons were based on exact 95% binomial confidence intervals. No differences in the care cascade were found except in the case of Cape Verde (sub-Saharan Africa, upper-middle-income group), where females in age groups 45–59 and 60–79 had significantly higher proportions in steps 1 and 2.

**Death rate.** In 100 countries (49%) mortality was significantly higher in males, in 9 countries (5%) mortality was significantly higher in females, and in 95 countries there was no significant difference in mortality rates.

### HIV and AIDS

**Risk factor exposure.** In 139 countries (68%) drug use prevalence was significantly higher among males in some age groups, whereas in Afghanistan, Belarus, China, Iceland, and Syria (3%) it was higher in females of some age groups. Unsafe sex prevalence was higher among females than males in 113 countries (55%), but the pattern was the opposite in Nepal at ages 30–34.

**HIV prevalence.** Significantly higher male prevalence rates of HIV were observed in 114 countries (56%), and 28 countries (14%) had higher female prevalence.

**Care cascade.** Proportions of diagnosed, treated, and controlled for HIV were available for ages 15+, with no age breakdown and for 76 countries: 16 high-income, 25 upper-middle-income, 28 lower-middle-income, and 7 low-income countries; 20 countries in Europe and Central Asia, 3 in South Asia, 21 in sub-Saharan Africa, 8 in Middle East and North Africa, 16 in Latin America and the Caribbean, 8 in East Asia and the Pacific, and no countries in North America. Females performed better in the care cascade in 9 (12%), 20 (26%), and 21 (28%) countries in diagnosis, treatment, and control steps, respectively. Conversely, better performance in the care cascade were observed among males in Lebanon for treatment and control.

**Death rate.** Higher male death rates from AIDS were observed in 131 countries (64%), and higher female death rates in 25 countries (12%).

## Discussion

Our analysis of sex-disaggregated data along the health pathway is a unique and ambitious attempt to holistically and systematically document sex differences in risk exposure, morbidity, care access, and mortality outcomes, based on existing and standardized datasets. The health pathway approach moves beyond focusing on a single step—whether it be one risk factor or death rate—to view (ill)health that is determined in continuation and accumulation along all steps. By examining the entire health pathway, we can identify the various points at which inequities arise, generate more nuanced hypotheses of the complex network of influences shaping health and mortality outcomes, and thereby design interventions that are more responsive to the roles that sex and gender are playing in outcome differences and arguably improve the health of all. We recognize that progression along the pathway is not inevitable—not every smoker will die from lung cancer, and not everyone who is obese will develop diabetes, for example. The pathway is also not designed to continuously model risks of progression from one step to the next. While acknowledging these shortcomings, this approach has revealed several insights for further exploration and action.

Findings of the study show significant sex differences in each step along pathways. In many countries, males exhibited higher disease prevalence and death rates than females, while in some countries, they also reported lower rates of healthcare seeking, diagnosis, and treatment adherence. For risk factors, males had significantly higher smoking rates in 86% of countries, whereas females had significantly higher smoking rates only in Bhutan. In contrast, females had significantly higher obesity rates in 64% of countries, while males had higher obesity rates only in Malawi and China, highlighting important differences in risk exposure, highlighting important differences in risk exposure [25]. Regarding disease prevalence, males had significantly higher prevalence rates of conditions in 56% of countries for HIV and AIDS, 30% for

diabetes, and 4% for hypertension. Females, on the other hand, had significantly higher prevalence rates of conditions in 14% of countries for HIV and AIDS, 5% for diabetes, and only in India for hypertension. Across all the three conditions, death rates were higher in males compared to females in 131 countries (64%) for HIV and AIDS, 107 for hypertension (53%) and 100 countries for diabetes (49%). In contrast, death rates were higher in females compared to males in 25 countries (12%) for HIV and AIDS, 9 countries for diabetes (4%), and only in the United Arab Emirates for hypertension. In the care cascades, females generally had higher rates of diagnosis, treatment, and control. This pattern was observed in 13% of countries for HIV and AIDS (10 countries), 2% for hypertension (3 countries), and only in Cape Verde for diabetes. Males had higher rates of HIV and AIDS treatment and control only in Lebanon. Such findings echo previous studies showing men's lower use of preventive, diagnostic, and treatment services compared to women, including for HIV, COVID-19, and other conditions [26–28]. Several factors, including rigid constructions of norms relating to masculinity [29], wider societal practices (such as clinic opening hours restricting men's access) [30], and structural drivers (including healthcare financing) [31] may be contributing to lower healthcare utilization among males. Improving population health, including through universal access to healthcare services, requires consideration of and action on barriers that restrict healthcare access and utilization for all people of all gender identities [32].

The analysis of the pathway data across the three conditions identified some opportunities for gender-responsive interventions to address some risk exposures. For example, smoking rates were higher in males almost worldwide, while obesity prevalence was higher among females in two-thirds of the countries. Such findings are not totally unexpected, but despite widespread and long-standing evidence of higher tobacco-smoking rates in men (and LGBTQ+ populations) [25], frequently associated with the socio-cultural construction of gender norms, and manipulated and promoted by the tobacco industry, relatively little work has been undertaken by the public health community to design gender-responsive tobacco control programmes. This represents a missed opportunity for improving population health and reducing gender gaps in health.

Previous studies have reported on the association between the cascade of care and factors associated with care progression across countries [6,13,17] but none of the studies discussed sex differences. In our study, while many countries showed significant sex differences in risk factors, disease prevalence, and death, fewer exhibited significant differences between males and females within the care cascade for hypertension, diabetes, and HIV/AIDS. For example, for HIV and AIDS, while almost 60% of the countries show significant sex differences in drug use, unsafe sex, disease prevalence, and death rates, only 13% of the countries report significant sex differences in diagnosis, treatment, and control. There are several potential explanations to these findings. We are only reviewing three conditions, which may happen to have smaller sex differences along the pathway than other leading causes of morbidity and mortality. Other conditions, such as mental health disorders, and infant and neonatal health [9,10,21,22], or cardiovascular diseases [33,34] have exhibited more pronounced sex-differences in care-seeking patterns in some countries. Once in the care system, studies have shown differences in the quality of care received— nationwide cohort study in England and Wales found that women suffering a heart attack were less likely to receive care that followed clinical guidelines [35]. Other studies have found higher mortality rates for female heart attack patients treated by male physicians, suggesting gender bias [36]. These differences illustrate the importance of having sex-disaggregated data along the care cascade for all conditions—findings will vary by condition and country, and the underlying reasons for inequities will vary, meaning that interventions need to be data-driven and context specific.

Another key finding of the study is a notable absence of globally available sex- and age-disaggregated care cascade data. We identified only three conditions, among the eight major conditions we explored, that have sex-disaggregated cascade data for more than five countries. The absence of this data for some major contributors to morbidity and mortality was surprising. For example, we were unable to identify global sex-disaggregated care cascade datasets for tuberculosis. A previous study reported that while males are biologically more vulnerable to tuberculosis than females, traditional gender norms may discourage women from using prevention and treatment services [37]. However, without sex-

disaggregated data on the tuberculosis cascade, it is difficult to identify who is left behind and where to intervene most strategically on the pathway. Although global data groups such as NCD RisC include data disaggregated by sex, studies analyzing trends in risk factors, prevalence, and treatment for the three conditions we reviewed often provide limited discussion on sex differences or focus primarily on aggregate populations without explicitly assessing sex-based disparities. While several studies report estimates separately for females and males, they mostly do not include a detailed assessment of sex differences in outcomes (e.g.,[6,38–40]). For example, a recent analysis of global trends in hypertension prevalence and treatment from 1990 to 2019 using data from the NCD Risk Factor Collaboration presented estimates stratified by sex but did not deeply examine sex differences in trends and outcomes [6,38]. Other studies, such as those by Rahim and colleagues [41] and Lee and colleagues [42], focus on total population estimates without sex-specific analyses. Additionally, studies have limitations in geographical coverage [39,42,43]. For example, three recent papers examined diabetes risk and preventive measures across low- and middle-income countries and the diagnostic testing for hypertension, diabetes, and hypercholesterolemia within similar economic groups [17,40,41], while another study explored trends in hypertension awareness, treatment, and control across 12 high-income countries [21]. Moreover, sex-disaggregated analyses, particularly regarding the care cascade for specific diseases, remain limited [11].

Among the identified datasets we used in this study, their coverage and completeness are far from satisfactory. While complete hypertension care cascade data was available for about 200 countries, for diabetes we were only able to construct the cascade for 39 countries in the last decade, and for most countries data were available for only one time period thus excluding the possibility of monitoring trends over time. Similarly, although data on the HIV and AIDS care cascade were available for 76 countries, these were not stratified by age. Other studies have found that age-specific data are essential for designing effective HIV and AIDS programmes that can adapt to the changing healthcare needs of populations as they age [14]. These limitations highlight the need for the creation of comprehensive and harmonized cascade reporting systems on a wider range of conditions, disaggregated by sex and age, with broader geographical and temporal coverage.

Among 204 countries, our analysis found significantly higher mortality rates in males compared to females in more than half of the countries. Our analysis cannot explain these differences (or similarities), but can highlight areas for further exploration with the lenses of both gender and sex. For example, are men subject to the constructions of masculinities that often discourage prevention and care-seeking, presenting later in the disease progression at health facilities compared to women (and hence more at risk of mortality from the diagnosed disease)? Are women's causes of death being under-reported or mis-attributed? Are there biological reasons why males have a higher mortality rate from diabetes compared to females—and, if so, does this warrant sex-specific clinical guidelines? We cannot address these questions within the pathway, but the pathway can pinpoint questions for deeper analysis and future research.

Despite a strong evidence base confirming the importance of sex and gender as determinants of disease, health policies often do not include sex-specific responsive approaches. Global Health 50/50 found that 91% of COVID-19 national health-related policies (of approximately 450 reviewed) did not consider gender [44]. The absence of gender-responsive policies in many areas of health results in gender-based inequities from risk exposure to care cascades that are not being adequately addressed in many settings. Acknowledging and addressing the unique health needs of women, men, and gender-diverse people through gender-responsive interventions is needed if we are to reduce health inequities across the whole population and better health outcomes for all.

Our study has some limitations including, the lack of available datasets and their own respective limitations. For example, on the diabetes and HIV cascade data, we observed variations in data quality and completeness across countries. Secondly, the data available may be affected by sample bias including the underrepresentation of certain populations, and inconsistencies in data collection methods. Reliance on national surveys may result in the exclusion or under-representation of some populations such as those marginalized by sexual or gender identity, or people living in geographically remote or in challenging conditions such as conflict zones. Self-reported data, inconsistencies in measurement

techniques and varying definitions of health conditions further complicate comparability. Thirdly, although we identify gender as determinant of inequities along the pathway, gender is expressed through identity, experienced through socio-cultural norms, and embedded in inequitable systems and structures [32]. Therefore, our ability to reduce health inequities will require acknowledging and addressing the role of gender across all these levels. Fourthly, we cannot empirically link the steps in the health pathway with each other. For example, we know that for any given condition the data on risk factor exposure and prevalence are related, but we cannot causally associate risk factor exposure and prevalence based on the datasets at hand, all of which were collected or modeled separately. This highlights the importance of longitudinal studies for a more nuanced understanding of causal relationships and sequential progress along the health pathway.

Based on our findings, we urge the collection and publication by government and research entities of comparable global data along the health pathway that is disaggregated by sex and age, and that it starts by focusing on the leading causes of death and morbidity. Without such data, we will be navigating the health pathway without clear insights into health inequities, thus limiting our ability to identify which population is suffering inequity (e.g., in rates of exposure to risk factors, low rates of access to health services, quality of care received, etc.) and thus where gender-responsive interventions might reduce inequities.

In order to tackle health inequities, we need more policy directives for data disaggregation across more intersectional characteristics and more health conditions and for the reporting of such data in the public domain. We call on policymakers to attempt to make greater use of such data to identify and respond to those population groups stumbling along or falling off the pathway. By integrating our understanding of the sex and gender differences through analyses such as this, policymakers can be encouraged to develop and implement gender-responsive strategies and interventions. Moreover, by comparing countries along the health pathways, our approach allowed us to identify countries that do not follow the general trends in sex differences, encouraging further investigation into the reasons behind the discrepancies. Disaggregating by other characteristics—urban/rural location, socio-economic position, race/ethnicity, disability, etc. —would add even greater specificity and permit even more effective policy and programme responses.

In addition to more disaggregated data, there is a need for inclusive data collection tools and harmonized reporting guidelines. This would enable more robust secondary analyses, including cross-country comparisons and in-depth exploration of subpopulations. Achieving this requires leadership from large international organizations or consortia, such as WHO or GBD, to ensure global alignment and coordination. To assemble and publish standardized datasets for additional health pathways, several options present themselves. As directing and coordinating authority for international cooperation on health, WHO can standardize data across countries, but it faces resource limitations and many competing priorities. Entities such as Global Health 50/50, Data 2× or Equal Measures (https://equalmeasures2030.org/members/data2x/) can seek private and other funding and operate independently, but they lack the convening power and generally must rely on publicly available data. Condition-specific epistemic communities offer deep expertise in their areas, yet they focus on one condition at a time, making comparisons challenging. Therefore, a collaborative approach might be the most effective solution to more joined up data coalition and analysis along condition-specific pathways. Finally, future iterations of country pathways should incorporate more upstream legal and policy environments, as these significantly influence risk exposures.

## Supporting information

**S1 Text.** **Table A.** List of countries with available data on risk factors, disease prevalence and death rate, and care cascade for hypertension, diabetes, and HIV and AIDS. **Fig A.** Sex differences (significant when non-overlapping confidence intervals of estimates between females and males) in health pathways of hypertension, by income group. **Fig B.** Sex differences (significant when non-overlapping confidence intervals of estimates between females and males) in health pathways of hypertension, by region. **Fig C.** Sex differences (significant when non-overlapping confidence intervals of estimates between females and males) in health pathways of diabetes, by

income group. Diabetes care cascade data are available only for age groups 30–44, 45–59, and 60–79. **Fig D.** Sex differences (significant when non-overlapping confidence intervals of estimates between females and males) in health pathways of diabetes, by region. Diabetes care cascade data are available only for age groups 30–44, 45–59, and 60–79. **Fig E.** Sex differences (significant when non-overlapping confidence intervals of estimates between females and males) in health pathways of HIV and AIDS, by income group. HIV and AIDS care cascade data are available only for the age group 15+ . **Fig F.** Sex differences (significant when non-overlapping confidence intervals of estimates between females and males) in health pathways of HIV and AIDS, by region. HIV and AIDS care cascade data are available only for the age group 15+ . **Fig G.** Percentages of countries with significant sex differences in global health pathways of hypertension, diabetes, and HIV and AIDS (significant when non-overlapping confidence intervals of estimates between females and males). **Fig H.** Percentages of countries with significant sex differences in health pathways of hypertension, by income group (significant when non-overlapping confidence intervals of estimates between females and males). **Fig I.** Percentages of countries with significant sex differences in health pathways of hypertension, by region (significant when non-overlapping confidence intervals of estimates between females and males). **Fig J.** Percentages of countries with significant sex differences in health pathways of diabetes, by income group (significant when non-overlapping confidence intervals of estimates between females and males). **Fig K.** Percentages of countries with significant sex differences in health pathways of diabetes, by region (significant when non-overlapping confidence intervals of estimates between females and males). **Fig L.** Percentages of countries with significant sex differences in health pathways of HIV and AIDS, by income group (significant when non-overlapping confidence intervals of estimates between females and males). **Fig M.** Percentages of countries with significant sex differences in health pathways of HIV and AIDS, by region (significant when non-overlapping confidence intervals of estimates between females and males).
(DOCX)

## Acknowledgments

All authors thank IHME for providing prevalence data and NCD-RisC for providing country-level data, and Sonja Tanaka for her administrative support.

## Author contributions

**Conceptualization:** Angela Y. Chang.

**Data curation:** Alessandro Feraldi, Virginia Zarulli, Angela Y. Chang.

**Formal analysis:** Alessandro Feraldi.

**Funding acquisition:** Kent Buse, Sarah Hawkes, Angela Y. Chang.

**Investigation:** Alessandro Feraldi, Virginia Zarulli, Angela Y. Chang.

**Methodology:** Alessandro Feraldi, Virginia Zarulli, Angela Y. Chang.

**Project administration:** Virginia Zarulli, Angela Y. Chang.

**Resources:** Virginia Zarulli, Angela Y. Chang.

**Software:** Alessandro Feraldi, Virginia Zarulli.

**Supervision:** Alessandro Feraldi, Virginia Zarulli, Kent Buse, Sarah Hawkes, Angela Y. Chang.

**Validation:** Alessandro Feraldi, Virginia Zarulli, Kent Buse, Sarah Hawkes, Angela Y. Chang.

**Visualization:** Alessandro Feraldi.

**Writing – original draft:** Alessandro Feraldi, Virginia Zarulli, Angela Y. Chang.

**Writing – review & editing:** Alessandro Feraldi, Virginia Zarulli, Kent Buse, Sarah Hawkes, Angela Y. Chang.

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
