## [Editor Report · Decision Letter 0]

17 Sep 2024

Dear Dr Feraldi,

Thank you for submitting your manuscript entitled "We Need More Sex-Disaggregated Data: Unveiling the Data Gap and Showing the Benefit of Sex-Disaggregated Data Along the Health Pathways for Hypertension, Diabetes, HIV and AIDS" for consideration by PLOS Medicine.

Your manuscript has now been evaluated by the PLOS Medicine editorial staff and I am writing to let you know that we would like to send your submission out for external peer review.

Please re-submit your manuscript within two working days, i.e. by Sep 19 2024 11:59PM.

Feel free to email me at pdodd@plos.org or the team at plosmedicine@plos.org if you have any queries relating to your submission.

Kind regards,

Pippa

Philippa C. Dodd, MBBS MRCP PhD

Senior Editor

PLOS Medicine

pdodd@plos.org

---

## [Decision Letter · Decision Letter 1]

20 Nov 2024

Dear Dr Feraldi,

Many thanks for submitting your manuscript "We Need More Sex-Disaggregated Data: Unveiling the Data Gap and Showing the Benefit of Sex-Disaggregated Data Along the Health Pathways for Hypertension, Diabetes, HIV and AIDS" (PMEDICINE-D-24-03084R1) to PLOS Medicine. The paper has been reviewed by subject experts and a statistician; their comments are included below and can also be accessed here: [LINK]

As you will see, the reviewers were generally positive about your paper, noting that it addresses a key gap in existing literature. However, there were a number of suggestions for improvement. After discussing the paper with the editorial team and an academic editor with relevant expertise, I'm pleased to invite you to revise the paper in response to the reviewers' comments. We plan to send the revised paper to some or all of the original reviewers, and we cannot provide any guarantees at this stage regarding publication.

We ask that you submit your revision by Dec 11 2024 11:59PM. However, if this deadline is not feasible, please contact us by email, and we can discuss a suitable alternative.

Best regards,

Syba

Syba Sunny

Associate Editor

PLOS Medicine

plosmedicine@plos.org

Comments from the academic editor:

The academic editor was supportive of the decision to issue a major revision outcome. However, he also had some comments. Two of his comments pertain to suggestions raised by some of the reviewers: (a) the need for more detail on what models were done and (b) the need for more detail on the sample selection issue. For (b), he noted that the different diseases use a different selection of countries, and this would likely have an impact on how findings could be compared across diseases, such as diabetes vs hypertension. He also commented that, given that GBD 2021 was released some months ago now, and, additionally, NCD RiSC recently released their diabetes prevalence and treatment paper, the authors may wish to incorporate these new estimates into their study. Finally, he added that the authors state their search of STEPS surveys since 2013 found n=39 countries – he wondered whether the authors could elaborate in their revised manuscript on why the search was restricted to 2013, and why it appears the search is some years out-of-date.

Comments from the reviewers:

Reviewer #1: This manuscript fills a key meta-data gap at the global level. It was very informative to see data presented on risk factors, disease prevalence and mortality for a range of diseases across most countries in the world. The sex and age disaggregation of data provide richer context and nuance to the findings.

Having said that, sometimes sex and gender are used interchangeably in the paper, which is confusing. Some specific examples and recommendations:

1. The title reads "We need more sex-disaggregated data…" but the first sentence of the abstract reads "Health data disaggregated by sex or gender identity is vital…".

Based on the discussion section in the paper, the authors presumably want to say in the abstract that health data disaggregated by sex AND gender are vital. And in the title, I recommend that the authors decide if they want the take home message of this manuscript to be that we need more sex-disaggregated data or whether we need more sex AND gender-disaggregated data. It is entirely appropriate for the authors to decide one way or the other, since both sex and gender disaggregated data are vital to understanding the world' pressing health problems (and for designing appropriate solutions to those problems). But right now, sex is referred to in some places, gender is mentioned in other places, and it is confusing when the terms are used interchangeably.

2. On page 11, there is a sentence that states "Despite these sex and gender differences, health policies often do not include sex- and gender-specific/responsive approaches." From the results, however, there are no gender differences presented because - as the authors point out - no adequate gender-disaggregated data were available in the data sources they examined. I recommend that the authors delete "gender" in the sentence. The authors have already explained in the discussion section that one key limitation of the data sources they examined was that they did not have adequate gender disaggregated data.

Reviewer #2: Thank you for this important analysis. I found the paper interesting, useful, and well done. Here are some points for consideration in both the framing and presentation of data that I think would be important to expand on in subsequent drafts.

1. For the other diseases for which global sex-disaggregated data were not available, was it that there were good global data, but it just wasn't disaggregated or that theses diseases lacked globally comparable data in general? It would be useful to clarify this, since there are different recommendations depending on the available data.

2. It is difficult to understand the author's statistical approach from the methods section alone, without reading the annexes. It would be useful to add some of the details from the annex into the last paragraph of the methods section. For example, detail your primary outcomes, "exposure", and covariates, how many models you built, variable selection for final models etc.

3. I note that the authors have presented stratified results by world region and income level, which is useful. Additionally, I think it would be interesting to examine the effect of country income as an effect modifier between sex and health outcomes. From reviewing the tables, it seems that women are experiencing poorer outcomes in sub-Saharan Africa and maybe other LICs while the inverse is true for men. When the results are presented just as global counts of countries where men and women are experiencing poorer health we may be missing a key part of the story - that country income level really matters for sex differences.

4. I think a more nuanced discussion of how sample bias may be impacting these results in warranted. For example, sex-disaggregated data may be more available in high-income countries (which may also be the countries where men are experiencing poorer health). I think a discussion about how global data availability, country-income level, and health disparities by sex/gender are interacting and impacting these results is warranted.

Reviewer #3: The manuscript presents a scoping review to identify gender and sex differences in the epidemiology (exposure, prevalence, care cascade and mortality) for several major health conditions. The authors identify several consistent sex-based differences for hypertension, diabetes and HIV/AIDS. The authors make a plea for wider availability of sex/gender- and age-disaggregated data. This is a very interesting manuscript with an important message. I think the manuscript can be improved by greater clarity on the data search and inclusion criteria and a clearer presentation with maybe a slightly deeper discussion of the results.

PURPOSE OF REVIEW

This is a statistical review and focuses foremost on statistical rigour of the study, though I do add some less statistical comments.

I note that this is not a statistics-heavy paper, so my review is relatively light-touch and I really only have 2 purely statistical comments (comments O.3 and M.3 in the comments below).

OVERALL ASSESSMENT

This is a very interesting study and I broadly agree with the authors' plea for more sex-disaggregated data being made available. I have several overall comments, with the most pertinent comment probably being that I do not think it is very clear at present hwo datsets were identified and selected for inclusion in / exclusion from the study (several of my comments touch upon this).

O.1: While the authors state that they searched "for published papers using global datasets on the sex-disaggregated care cascade", yet only used data from the Global Burden of Disease (GBD) 2019. For the care cascade 3 sources were used (NCD-RisC, STEPS and UNAIDS). I have 2 questions regarding the use of GBP 2019:

1) as far as I know GBD 2021 data are available, why were these not used?

2) While GBD data are fairly comprehensive and definitely global, there are datsets outside of GBD. I would expect that some of the data that the authors highlight as not being available, are probably available on a country-by-country basis, with varying data access requirements and varying data quality (e.g. from routinely collected data, from surveys, ...). I do realise that it is highly impractical to navigate all of these issues for the kind of study presented here and the authors are transparent that they only considered "global" datasets, but I do not think the paper currently is very clear how datsets were searched for and what the exact criteria were for inclusion in the study. Can the authors add details of other datasets that were considered but ultimately decided not to be used (and for what reason)? This would also be useful to understand the dataset identification and selection for the care-cascade analysis.

If it was an active decision to only use GBD data, then I think this should be very clearly stated and acknowledged as a limitation.

O.2: I am not sure the authors go far enough with their plea. We do not just need more data, but also standardised data collection tools, analysis frameworks and reporting guidelines to make it easier to conduct secondary, cross-country analyses that can drill down into sub-populations etc. This would have to be lead by large international bodies or consortia (e.g. WHO, GBD, ...).

O.3: The results as presented in this manuscript rely on the definition of statistical significance used by the authors. I do not think this definition is optimal. On p.6 at the top, the authors state "We determined sex differences by comparing the confidence intervals between females and males, and considered steps with non-overlapping confidence intervals as statistically significant." While technically this statement cannot be ruled to be incorrect (since it simply states a definition), what this statement implicitly suggests is this: if the 95% CIs for the estimates of the same parameter in 2 different groups do not overlap, then that difference in parameter estimates is statistically significant at the 5% significance level. That statement is not fully correct: the difference willc ertainly be significant at the 5% level, but it will also be significant at a much lower level. In fact, a parameter, estimated in 2 groups, can have a statistically significant (at the 5% level) difference between these groups yet the 95% confidence intervals for the 2 groups overlap. Generally a more valid rule would be to consider statistical significance if the point estimate of one group is not contained within the 95% CI of the other group. The current definition of statistical significance would be considerably lower than the commonly used 5% level (and while that 5% is arbitrary, it is commonly used and that is what most readers of the manuscript would take that your statistical significance implies).

INTRODUCTION

(No comments from me on this section.)

METHODS

M.1: Unclear how the search was conducted before results were filtered down to those conditions for which disaggregated datasets for at least 5 countries were available. Also what was the range of publication dates for inclusion in this review?

Would be good to have a supplement giving the exact search criteria so that the search could be reproduced.

M.2: Was the search for data limited to Global Burden of Disease? For most countries, there would be more data available from in-country sources. Did the authors try to access such data (I admit this would probably be a fairly mammoth task)?

M.3: p.6 at the top: The authors state that they calculated "asymptotic 95% binomial confidence intervals". In a frequentist analysis, all CIs are of course asymptotic. However, for a binomial CIs, these are more usually referred to as 'exact' CIs to contrast this with normal approximation based CIs for proportions (which in addition to the usual frequentist asymptotic interpretation of CIs, is further making use of the asymptotic distribution convergence to a normal distribution as dicdated by the Central Limit Theorem).

M.4: The methods section mentions multivariable regression models being used in hypertension and HIV to "identify socio-economic differences at country level and risk factors influencing sex differences". However the results section does not seem to report results from these analyses and also the discussion does not touch upon this. Please either remove this bit from methods if those regression analysis results will not be included or include and discuss those results.

RESULTS

R.1: Figure 2: Several comments for this figure: 1) clearer if this was labelled with panel labels (e.g. A, B, C, D, E, F), 2) colour scales are difficult to read - could you use palettes with higher contrast?, 3) for diabetes, I think the cascade can be shown on the same graphs -- yes the age groups are not the same but the boundaries are shared (i.e. for the cascade you can simply merge the cells for each set of 3 finer-scale age groups into a single cell.

R.2: My main comment on the result shown here is that it seems to be mostly an illustrative analysis of what widely-available sex/gender- and age-disaggregated data would allow to do. The authors mainly present results but do not dig much deeper in any of the things they identify. E.g. the authors state "Moreover, by comparing countries along the health pathways, our approach allowed us to identify countries that do not follow the general trends in sex differences, allowing us to probe further into the reasons behind the discrepancies." but then do not do this additional probing in the present manuscript. Likewise, while the authors discuss potential hypotheses that could explain the observed sex/gender differences, they do not pursue any of these further. I think there could be scope here for going a bit further with what the analysis has found. This could be where one would need to go from global datasets to more country-specific datasets.

R.3: This is not a very useful comment for the authors as I cannot think of a good suggestion for improving the presentation of results, but I still just want to state that since the authors present data on 3 conditions, each with multiple risk factors, and their care cascades, I find the results a bit difficult to navigate and getting a good sense of. Figure 2 and Tables 1-3 go a long way, but overall I am not sure I really 'got' the results.

DISCUSSION

(No comments from me on this section.)

REPORTING

RP.1: The authors identify the study as a scoping review, but did not indicate that they followed scoping review reporting guidelines. The quality and usefulness of the manuscript can be increased by following PRISMA scoping review extension guidelines (https://www.equator-network.org/reporting-guidelines/prisma-scr/).

OTHER COMMENTS

OC.1: The last word ("age group") from the 1st paragraph in the results section this should be in the plural, i.e. "age groups".

OC.2: p.9 "this data" -- more common to consider 'data' to be a plural and hence it should be "these data". Please also check the rest of the manuscript for consistent plural use of 'data'.

OC.3: p.9 "one-time period" should be "one time period".

---

* Please upload any figures associated with your paper as individual TIF or EPS files with 300dpi resolution at resubmission; please read our figure guidelines for more information on our requirements: http://journals.plos.org/plosmedicine/s/figures. While revising your submission, please upload your figure files to the PACE digital diagnostic tool, https://pacev2.apexcovantage.com/. PACE helps ensure that figures meet PLOS requirements. To use PACE, you must first register as a user. Then, login and navigate to the UPLOAD tab, where you will find detailed instructions on how to use the tool. If you encounter any issues or have any questions when using PACE, please email us at PLOSMedicine@plos.org.

FIGURES AND TABLES

SUPPLEMENTARY MATERIAL

* Please note that supplementary material will be posted as supplied by the authors.

REFERENCES

---

## [Decision Letter · Decision Letter 2]

28 Jan 2025

Dear Dr Chang,

Many thanks for submitting your revised manuscript "We Need More Sex-Disaggregated Data: Unveiling the Data Gap and Showing the Benefit of Sex-Disaggregated Data Along the Health Pathways for Hypertension, Diabetes, HIV and AIDS" (PMEDICINE-D-24-03084R2). The paper has been re-reviewed by the original reviewers; their comments are included below and can also be accessed here: [LINK]

As you will see, two of the reviewers were satisfied with the revised paper and your responses to their comments, and the statistical reviewer raised only a few minor points to be addressed. However, the academic editor raised some more fundamental questions about the framing of the paper, and the editorial team agrees that it will be important to consider these points carefully and revised the manuscript accordingly. As such, I'm pleased to invite you to revise the paper in response to the remaining reviewer and editorial comments. We may send the revised paper back to the statistician, and we will discuss the presentational changes with the academic editor. At this stage, we cannot provide any guarantees at this stage regarding publication.

We ask that you submit your revision by February 4th. However, if this deadline is not feasible, please contact me by email (hvanepps@plos.org), and we can discuss a suitable alternative. Please also feel free to contact me directly with any questions; otherwise, we look forward to receiving your revision.

Kind regards,

Heather

Heather Van Epps, PhD

Executive Editor

[on behalf of]

Philippa Dodd, MBBS MRCP PhD

Associate Editor

PLOS Medicine

pdodd@plos.org

Comments from the academic editor:

To me, this paper reads as though the authors had an a priori assumption that there is a paucity of sex-disaggregation, and this is the conclusion they were striving for. I think this is bias reflected not only in the title ("We Need More Sex-Disaggregated Data" -- not very subtle!) but also a lot in the discussion and what this study adds.

For example, the authors strenuously and repeatedly argue that other global data groups such as NCD RisC and HPACC (disclaimer: I am a member of both) "did not include assessment of sex differences" or "only a few include sex-disaggregated data." These statements are overstating their case. NCD RisC generated Bayesian models separately by sex in all of their papers. HPACC always includes sex/gender as a variable in individual regression models.

Another point is that in the discussion, the authors basically just go right into their talking points on how more disaggregated is needed, and the findings from the paper (which are legitimately interesting!) are basically buried.

For example, in the abstract, the authors write, "Significant sex differences were found in each step along the pathways for many countries. Compared to females, males exhibited higher disease prevalence and death rates and lower rates of health care seeking, diagnosis and treatment adherence in many countries."

Yet in the discussion, they write, "Discussion: "In our study, we found few countries with significant differences between males and females at any one of the points on the cascade for each of the three conditions."

So which is it? Again, I suggest 2 key points:

1. The authors in the first paragraph need to summarize the primary findings of the analysis. They can include an argument about the need for sex-specific data but it needs to be commensurate with the true findings from the paper.

2. Even more critically, the authors need to meaningfully address the fact that their data mostly support that men are overburdened and underserved by health care systems. Perhaps this is not the finding they had expected to find, perhaps it is politically inconvenient, and perhaps it goes against some of their group's intended advocacy. But to me it absolutely jumps off the page. Yet, it is kind of buried. To me, the main finding of this paper is that men are an underserved population. This is not to say that women should be ignored, but this key finding that should be front and center and that the authors need to meaningfully discuss.

The panels in Figure 2 are very difficult to interpret. They are inscrutable. I would suggest a key change. Have a single figure by disease and then have a spectrum shading that ranges from more countries male>female on one end to more countries female>male on the other end. Grey could be the neutral shade, rather than having 0% as a floor. I know the authors have an a priori argument about the need for sex-disaggregated cascades, but separating this figure does not help readers or further their argument. For example, look at the hypertension panels (Panels A and B) and the overweight column. It is yellow in both. What information does that convey? It means that sex differences are essentially balanced across the dataset, which is what the reader will intuitively be thinking. The figures could be pushed to the appendix or placed next to this new figure, which would thus illustrate both the between-country average and distribution.

With respect to the regressions, I am fine with them leaving them out. I kind of agree that presenting the unadjusted estimates is more in line with the goal of the paper.

Comments from the reviewers:

Reviewer #2:

The authors have adequately addressed all of my comments

Reviewer #3 (statistics):

I thank the reviewers for comprehensively responding to my queries. My concerns have largely been addressed.

I have 3 outstanding comments:

1. Statistical significance definition. I thank the reviewers for clarifying the implications of their methodology (i.e. difference being significant at a much lower level than the typical 5% that casual readers may have assumed). The implication that p-values for the differences are <=0.0025 is largely correct, but only if a few further assumptions (e.g. independence of the 2 estimators) are met.

2. Given the really stringent definition of significance, I think it is key in the results and discussion sections to not equate a lack of significance to a lack of a difference. For example the authors state that "Male smoking rates surpass those of females in 174 (87%) countries" but what they really mean is that smoking rates among men are significantly (according to the definition used in this manuscript) higher those among women in these countries. There may be other countries where these rates are higher, but not significantly so. I would suggest that the authors carefully review and revise the language used throughout the results and discussion sections.

3. The graphs are much easier to read now - thank you. The editors may want to check whether specific colours need to be changed for the benefit of colour-blind readers.

1. We ask every co-author listed on the manuscript to fill in a contributing author statement, making sure to declare all competing interests. If any of the co-authors have not filled in the statement, we will remind them to do so when the paper is revised. If all statements are not completed in a timely fashion this could hold up the re-review process. If new competing interests are declared later in the revision process, this may also hold up the submission. Should there be a problem getting one of your co-authors to fill in a statement we will be in contact. Please do not add or remove authors without first discussing this with the handling editor. You can see our competing interests policy here: http://journals.plos.org/plosmedicine/s/competing-interests.

2. *Please upload any figures associated with your paper as individual TIF or EPS files with 300dpi resolution at resubmission; please read our figure guidelines for more information on our requirements: http://journals.plos.org/plosmedicine/s/figures. While revising your submission, please upload your figure files to the PACE digital diagnostic tool, https://pacev2.apexcovantage.com/. PACE helps ensure that figures meet PLOS requirements. To use PACE, you must first register as a user. Then, login and navigate to the UPLOAD tab, where you will find detailed instructions on how to use the tool. If you encounter any issues or have any questions when using PACE, please email us at PLOSMedicine@plos.org.

3. Please ensure that the paper adheres to the PLOS Data Availability Policy (see http://journals.plos.org/plosmedicine/s/data-availability), which requires that all data underlying the study's findings be provided in a repository or as Supporting Information. For data residing with a third party, authors are required to provide instructions with contact information (web or email address) for obtaining the data. Please note that a study author cannot be the contact person for the data. PLOS journals do not allow statements supported by "data not shown" or "unpublished results." For such statements, authors must provide supporting data or cite public sources that include it.

4. We expect all researchers with submissions to PLOS in which author-generated code underpins the findings in the manuscript to make all author-generated code available without restrictions upon publication of the work. In cases where code is central to the manuscript, we may require the code to be made available as a condition of publication. Authors are responsible for ensuring that the code is reusable and well documented. Please make any custom code available, either as part of your data deposition or as a supplementary file. Please add a sentence to your data availability statement regarding any code used in the study, e.g. "The code used in the analysis is available from Github [URL] and archived in Zenodo [DOI link]" Please review our guidelines at https://journals.plos.org/plosmedicine/s/materials-software-and-code-sharing and ensure that your code is shared in a way that follows best practice and facilitates reproducibility and reuse. Because Github depositions can be readily changed or deleted, we encourage you to make a permanent DOI'd copy (e.g. in Zenodo) and provide the URL.

FORMATTING – GENERAL

5. Title: Please revised the title to adhere to PLOS convention. The title should be non-declamatory and should include a colon followed by the study design at the end of the title.

6. Abstract: Please structure your abstract using the PLOS Medicine headings (Background, Methods and Findings, Conclusions). Please combine the Methods and Findings sections into one section.

7. At this stage, we ask that you include a short, non-technical Author Summary of your research to make findings accessible to a wide audience that includes both scientists and non-scientists. The Author Summary should immediately follow the Abstract in your revised manuscript. This text is subject to editorial change and should be distinct from the scientific abstract. Ideally each sub-heading should contain 2-3 single sentence, concise bullet points containing the most salient points from your study. In the final bullet point of 'What Do These Findings Mean?', please include the main limitations of the study in non-technical language. Please see our author guidelines for more information: https://journals.plos.org/plosmedicine/s/revising-your-manuscript#loc-author-summary.

8. Please express the main results with 95% CIs as well as p values. When reporting p values please report as p<0.001 and where higher as the exact p value p=0.002, for example. Throughout, suggest reporting statistical information as follows to improve clarity for the reader "22% (95% CI [13%,28%]; p</=)". Please be sure to define all numerical values at first use.

9. Please include page numbers and line numbers in the manuscript file. Use continuous line numbers (do not restart the numbering on each page).

10. Please cite the reference numbers in square brackets. Citations should precede punctuation.

FIGURES AND TABLES

11. Please provide titles and legends for all figures and tables (including those in Supporting Information files).

12. Please define all abbreviations used in each figure/table (including those in Supporting Information files).

13. Please consider avoiding the use of red and green in order to make your figure more accessible to those with color blindness.

SUPPLEMENTARY MATERIAL

14. Please note that supplementary material will be posted as supplied by the authors. Therefore, please amend it according to the relevant comments outlined here.

15. Please cite your Supporting Information as outlined here: https://journals.plos.org/plosmedicine/s/supporting-information

REFERENCES

16. PLOS uses the numbered citation (citation-sequence) method and first six authors, et al.

17. Please ensure that journal name abbreviations match those found in the National Center for Biotechnology Information (NCBI) databases (http://www.ncbi.nlm.nih.gov/nlmcatalog/journals), and are appropriately formatted and capitalised.

18. Where website addresses are cited, please include the complete URL and specify the date of access (e.g. [accessed: 12/06/2023]).

19. Please also see https://journals.plos.org/plosmedicine/s/submission-guidelines#loc-references for further details on reference formatting.

---

## [Decision Letter · Decision Letter 3]

14 Mar 2025

Dear Dr. Chang,

Thank you very much for re-submitting your manuscript "Sex-Disaggregated Data Along the Gendered Health Pathways: A Review and Analysis of Global Data on Hypertension, Diabetes, HIV, and AIDS" (PMEDICINE-D-24-03084R3) for review by PLOS Medicine.

I have discussed the paper with my colleagues and the academic editor and it was also seen again by one of the reviewers. I am pleased to say that provided the remaining editorial and production issues are dealt with we are planning to accept the paper for publication in the journal.

[LINK]

We look forward to receiving the revised manuscript by Mar 21 2025 11:59PM.   

Sincerely,

Rebecca Kirk

On behalf of:

Suzanne De Bruijn, PhD

Senior Editor 

PLOS Medicine

plosmedicine.org

Requests from Editors:

GENERAL EDITORIAL REQUESTS

* Please confirm that your title complies with to PLOS Medicine's style. Your title must be nondeclarative and not a question. It should begin with main concept if possible. "Effect of" should be used only if causality can be inferred, i.e., for an RCT. Please place the study design ("A randomized controlled trial," "A retrospective study," "A modelling study," etc.) in the subtitle (ie, after a colon).

* Please confirm that your abstract complies with our requirements, including providing all the information relevant to this study type https://journals.plos.org/plosmedicine/s/submission-guidelines#loc-abstract

* Please ensure that the Introduction ends with a clear description of the study question or hypothesis.

* Please ensure that all abbreviations are defined at first use throughout the text.

GENERAL

* Please review your text for claims of novelty or primacy (e.g. 'for the first time') and remove this language. In addition, please check that any use of statistical terms (such as trend or significant) are supported by the data, and if not please remove them.

FUNDING STATEMENT

* The funding statement should include: specific grant numbers, initials of authors who received each award, URLs to sponsors’ websites. Also, please state whether any sponsors or funders (other than the named authors) played any role in study design, data collection and analysis, the decision to publish, or preparation of the manuscript. If they had no role in the research, include this sentence: “The funders had no role in study design, data collection and analysis, decision to publish, or preparation of the manuscript.”

COMPETING INTERESTS STATEMENT

* All authors must declare their relevant competing interests per the PLOS policy, which can be seen here: https://journals.plos.org/plosmedicine/s/competing-interests For authors with ties to industry, please indicate whether any of the interests has a financial stake in the results of the current study.

ETHICS AND CONSENT

* Please review your manuscript and edit to ensure compliance with our inclusive language requirements https://journals.plos.org/plosmedicine/s/human-subjects-research#loc-categorization

Comments from Reviewers:

Reviewer #3: I thank the reviewers for comprehensively responding to my outstanding queries. My original queries are all resolved now.

However, I have a new point of query which results from an apparent change in the results / data in this latest revision. Suddenly in this revision there are data for 204 rather than 200 countries, with some results also slightly changed (e.g. the smoking rates significantly surpassing those of females in 176 compared to previously 174 countries). There does not appear to be an explanation for this change in data / results in the reviewer/editor response letter and it does not appear to be due to a request/query from another reviewer. Maybe I missed it. Could you please elaborate why this change? It appears to be a minor change, but important to understand when data underlying a manuscript change from oen revision to the next.

[LINK]

---

## [Editor Report · Decision Letter 4]

28 Mar 2025

Dear Dr Chang, 

On behalf of my colleagues and the Academic Editor, David Flood, I am pleased to inform you that we have agreed to publish your manuscript "Sex-Disaggregated Data Along the Gendered Health Pathways: A Review and Analysis of Global Data on Hypertension, Diabetes, HIV, and AIDS" (PMEDICINE-D-24-03084R4) in PLOS Medicine.

PRESS

Sincerely, 

Rebecca Kirk

On behalf of

Suzanne De Bruijn, PhD 

Senior Editor 

PLOS Medicine